# MALDI-TOF: A new tool for the identification of *Schistosoma* cercariae and detection of hybrids

Antoine Huguenin[1,2]*, Julien Kincaid-Smith[3,4], Jérôme Depaquit[1,2], Jérôme Boissier[3], Hubert Ferté[1]

**1** Université de Reims Champagne Ardenne, EA7510 ESCAPE, Reims, France, **2** Laboratoire de Parasitologie-Mycologie, pôle de Biopathologie, CHU de Reims, Reims, France, **3** IHPE, Université de Montpellier, CNRS, Ifremer, Université de Perpignan, Perpignan, France, **4** CBGP, IRD, CIRAD, INRAE, Institut Agro, Université de Montpellier, Montpellier, France

* ahuguenin@chu-reims.fr

**Data Availability Statement:** All relevant data are within the manuscript and in its supporting information files. The MSP database is accessible

## Abstract

Schistosomiasis is a neglected water-born parasitic disease caused by *Schistosoma* affecting more than 200 million people. Introgressive hybridization is common among these parasites and raises issues concerning their zoonotic transmission. Morphological identification of *Schistosoma* cercariae is difficult and does not permit hybrids detection. Our objective was to assess the performance of MALDI-TOF (Matrix Assisted Laser Desorption-Ionization–Time Of Flight) mass spectrometry for the specific identification of cercariae in human and non-human *Schistosoma* and for the detection of hybridization between *S. bovis* and *S. haematobium*. Spectra were collected from laboratory reared molluscs infested with strains of *S. haematobium*, *S. mansoni*, S. *bovis*, *S. rodhaini* and *S. bovis* x *S. haematobium* natural (Corsican hybrid) and artificial hybrids. Cluster analysis showed a clear separation between *S. haematobium*, *S. bovis*, *S. mansoni* and *S. rodhaini*. Corsican hybrids are classified with those of the parental strain of *S. haematobium* whereas other hybrids formed a distinct cluster. In blind test analysis the developed MALDI-TOF spectral database permits identification of *Schistosoma* cercariae with high accuracy (94%) and good specificity (*S. bovis*: 99.59%, *S. haematobium* 99.56%, *S. mansoni* and *S. rodhaini*: 100%). Most misidentifications were between *S. haematobium* and the Corsican hybrids. The use of machine learning permits to improve the discrimination between these last two taxa, with accuracy, F1 score and Sensitivity/Specificity > 97%. In multivariate analysis the factors associated with obtaining a valid identification score (> 1.7) were absence of ethanol preservation (p < 0.001) and a number of 2–3 cercariae deposited per well (p < 0.001). Also, spectra acquired from *S. mansoni* cercariae are more likely to obtain a valid identification score than those acquired from *S. haematobium* (p<0.001). MALDI-TOF is a reliable technique for high-throughput identification of *Schistosoma* cercariae of medical and veterinary importance and could be useful for field survey in endemic areas.

at https://zenodo.org/ under the following DOI: 10.5281/zenodo.6631761.

**Funding:** Funding received by JB This research was funded by the program HySWARM (ANR18-CE35-0001) from the French Research National Agency. https://anr.fr/Projet-ANR18-CE35-0001 The funders had no role in study design, data collection and analysis, decision to publish, or preparation of the manuscript.

## Author summary

Schistosomiases are neglected tropical diseases, affecting approximately 200 million people worldwide. They are transmitted during contact with water contaminated with the infesting stage of the parasite (the cercaria stage). Species-level recognition of cercariae present in water has important implications for field campaigns aimed at eradicating schistosomiasis. In addition, Schistosomes are able to hybridize between different species. Identification of Schistosomes cercariae on microscopy is difficult because of their similarity, and it does not allow hybrids to be distinguished. Molecular biology techniques allow a reliable diagnosis but are expensive. MALDI-TOF is a recent technique that permits an inexpensive identification of micro-organisms in a few minutes. In this paper, we evaluate MALDI-TOF identification of Schistosomes cercariae.

We have implemented a database of MALDI-TOF cercariae spectra obtained from parental strains and hybrids of species of medical or veterinary interest, allowing reliable identification with an accuracy of 94%. The identification errors mainly come from confusion between the natural Corsican hybrid (*S. haematobium* x *S. bovis*) and *S. haematobium*. The use of machine learning algorithms permits to obtain an accuracy of more than 97% in the recognition of these two parasites. In conclusion, MALDI-TOF is a promising tool for the identification of Schistosome cercariae.

## Introduction

Schistosomiasis is a neglected water-born parasitic disease caused by digenean Trematoda of the genus *Schistosoma*. These parasites have a complex life cycle involving a fresh-water mollusc as an intermediate host for asexual reproduction. As a result of this process, a free-swimming larval stage, the cercariae, are released from infected snails in water which can penetrate the skin of the bathing definitive hosts [1,2].

By mid-2003, the number of people at risk of developing schistosomiasis was estimated at 779 million (more than 10% of the world population) [3]. The greatest burden of schistosomiasis is observed in Sub-Saharan Africa [4], in which *S. haematobium* and *S. mansoni* are the most common parasites [2]. In 2013 more than 200 million people were estimated to be infected [5]. An integrated approach has been proposed by WHO for eradication of schistosomiasis involving periodic mass treatments with preventive anti-helmintic chemotherapy, water sanitation and hygiene (WASH strategy) and snail control [6]. Morbidity due to schistosomiasis has recently declined, going from 2,543,364 disability adjusted life years (DALYs) in 2016 to 1,627,844 in 2019 [7]. The WHO goal for 2030 is to eliminate schistosomiasis as a public health problem in all 78 endemic countries [8].

Precision mapping of *Schistosoma* prevalence in man and in the environment is thus necessary for achievement of this goal [9]. Detection of *Schistosoma* infection in its snail host needs efficient tools to monitor the efficacy of elimination programs [1]. Snails could also serve as "sentinels" to evaluate the human risk of infection as a relation between snail infection and local human genotypes near the surveyed snail habitats has been reported [10].

A tool permitting to screen rapidly harvested snails for infection with human pathogenic *Schistosoma* would be particularly useful to achieve this task. This efficient xenomonitoring could also allow more precisely targeted mass chemotherapy administration campaigns increasing the efficiency of the control.

The induction of *cercariae* released by light stimulation allows the identification of a patent snail infection. Such time-consuming methods require, however, an experienced microscopist

in light of the difficulty to identify cercariae species using conventional morphological criteria. Indeed, livestock, wildlife and human cercariae are often co-endemic and are not easily distinguishable by conventional microscopy [11]. Only a precise analyse of papillae distribution on cercariae surface (i.e. Chaetotaxy) allows to distinguish these *larvae* at a species level [12]. Various molecular technics have been developed to achieve this task [13–16] but they remain expensive and time-consuming.

Another level of complexity is due to introgressive hybridization between *Schistosoma* species. Indeed, several hybrid schistosomes have been identified in the field between human infecting species (e.g. *S. haematobium x S. mansoni*), between animal infecting species (e.g. *S. bovis x S. curassoni*) and between human and animal infecting species (e.g. *S. bovis x S. haematobium*) [17]. These latter hybrids are certainly the most worrying because they raise the eventual capability of zoonotic transmission as evidenced in Benin previously [18,19]. Among hybrid schistosomes, *S. haematobium x S. bovis* hybrids received the most attention and have been identified in several West African countries including Sénégal, Benin, Côte d'Ivoire, Cameroon, Nigeria, Mali and Niger [17,20–22]. These hybrids have also been responsible of an outbreak in Corsica [12], probably originating from a human migrant with autochtonous transmission by local *Bulinus truncatus* snails. As a consequence of hybridization, hybrid vigor and adaptive introgression in schistosome populations may lead to parasites producing more eggs with a larger size than their parental form, and thus expand their intermediate and definitive host spectrum [23]. Identification of Schistosome species and their hybrids is thus of particular interest to better control parasites' transmission and morbidity.

MALDI-TOF (Matrix Assisted Laser Desorption-Ionization–Time Of Flight) mass spectrometry, an important tool in microbiology, already allows the identification of bacteria, fungi arthropods or protozoans [24–26]. In helminthology, MALDI-TOF has been used for the identification of various nematodes and Platyhelminthes [27]. It has also recently been used successfully to identify cercariae of European Trematoda [28]. In *Schistosoma*, Hamlili et al. have demonstrated the usefulness of MALDI-TOF in malacology to identify *Schistosoma*'s intermediate hosts [29]. The authors, however, were not able to detect *Schistosoma* signals in the spectra of snail's tissues in order to detect non patent infections. Several papers have also been published on the use of MALDI-TOF for detecting circulating biomarkers of *S. japonicum*. First in a rabbit model [30] then in mice used as sentinel [31] and finally for differentiating patients with newly developed advanced schistosomiasis from healthy controls [32].

Our objective was to assess the performance of MALDI-TOF for the specific identification of cercariae in human and non-human *Schistosoma* and for the detection of hybridization between *S. bovis* and *S. haematobium*.

## Material and methods

### Ethics statement

Experiments were carried out according to national ethical standards of the French guidelines (writ of February 1st, 2013; NOR: AGRG1238753A). Experiments were carried out under the permit A66040 delivered by French Ministry of Food and Agriculture. The investigator possesses an official certificate for animal experimentation (number of the authorization 007083).

### Strains

*Schistosoma rodhaini*, *S. mansoni*, *S. bovis*, *S. haematobium* and *S. haematobium-bovis* hybrid strains were used in this study.

The *S. bovis* strain was isolated in 1970 in Villar de la Yegua-Salamanca (Spain) and was kindly provided by Ana Oleaga from the Spanish laboratory of parasitology of the Institute of

Natural Resources and Agrobiology in Salamanca. This strain is maintained in golden hamster (*Mesocricetus auratus*) as definitive host and *Bulinus truncatus* (Spanish strain) or *Planorbarius metidjensis* (Spanish strain) as intermediate hosts.

*S. haematobium* was originally isolated circa 1950 from an unknown location in Egypt. The laboratory stock of the Egyptian strain of *S. haematobium* was later mixed with an isolate that was thought to be obtained from Abrawash (Cairo) by the Naval Medical Research Unit III, in 1977. The current Egyptian strain of *S. haematobium* that is maintained at the Biomedical Research Institute (Rockville, USA) is from a mixture of the 1977 stock with another Egyptian isolate obtained in the 1980s. This strain was kindly provided by the BRI. Although used as *S. haematobium's* reference genome [33], recent genomic characterization of this strain suggests that in fact it contains introgressed *S. bovis* alleles in genomic regions spanning up to 100 kb [34].

The *S. mansoni* strain was isolated from an infected patient in Recife Hospital (Brazil). The strain was installed in the lab in 1975 after a generous gift of Pr Y. Golvan (Hôpital Saint Antoine, Paris).

*S. rodhaini* was isolated from an infected rodent in Burundi. The strain was installed in the lab in 1984 after a generous gift of Pr D. Rollinson (British Museum, London).

The *S. haematobium-bovis* hybrid strain was recovered from eggs isolated from a patient infected in the Cavu River, Corsica, France [35]. Since 2014 this strain is maintained in golden hamster as definitive host and *Bulinus truncatus* (Corsican strain) as intermediate host. This strain was genetically characterized at the whole genome level and is composed of 77% and 23% of *S. haematobium* and *S. bovis*, respectively [35].

*S. haematobium x S. bovis* reciprocal first generation hybrids (e.g. F1 ou F1') were obtained by crossing the two parental strains according to a previously described protocol [36]. Briefly, molluscs are individually exposed to a single miracidia in order to obtain single-sex clonal populations of *cercariae*. Secondly, cercarial populations are molecularly sexed [37]. Thirdly, infected molluscs are gathered and separated according to the species and sex of the schistosome infection: *B. truncatus* infected by male or female, *S. bovis* or *S. haematobium*. Lastly, hamsters are exposed to equal numbers of male *S. bovis cercariae* and female *S. haematobium* cercariae (F1) or the reciprocal combination (equal numbers of female *S. bovis* and male *S. haematobium*: F1').

## Spectra acquisition

Cercariae emissions and MALDI-ToF spectra were obtained as previously described by Huguenin et al. [28]. Briefly, snails were isolated individually and cercarial emergence was obtained by light stimulation lasting between 30 min and 2h. 2–5µL of water was then deposited in each spot on a 96 wells polished steel MALDI-ToF target plate (Bruker Daltonics GmbH, Bremen, Germany).

The abundance of emitted cercariae in water and the number of deposited cercariae per spot was assessed under stereomicroscope. The number of replicates per plates and the number of deposited cercariae per well was dependent on the abundance of emitted cercariae.

After complete drying, 1 µL of formic acid (Sigma Aldrich, Saint Quentin Fallavier France) was added to each well and after evaporation spots were covered by 1 µL of MALDI HCCA matrix (α-cyano-hydroxy-cinnamic acid in solution with 2.5% trifluoroacetic acid and 50% acetonitrile in water, Bruker Daltonics). Spectra were acquired with a Microflex LT mass-spectrometer (Bruker Daltonics), using default parameters (linear positive ions mode acquired on a range of 2000–20,000 Da). In order to produce a sufficient amount of spectra each spot was read at least 8 times. All spectra were acquired using the standard

MBT_BTS_Validation_AutoX method using FlexControl v3.4, except one plate inadvertently acquired using MBT_AutoX_smart.

## Spectra analysis

**Spectral quality and reproducibility assessment.** Spectra were visually assessed in FlexAnalysis v3.4 and imported in Biotyper Compass Explorer v4.1.100 for analysis.

The assessment of spectral intraspecific reproducibility and of its variation factors was performed by computing the Composite Correlation Index (CCI) using Compass Explorer default settings with the following spectral features: (i) sum of the intensity of all detected peaks, using R scripts adapted from Cuénod et *al.* [38] and (ii) signal to noise ratio (SNR) of the spectra calculated by the MALDIQuant detectPeaks function. PCA analyse were performed using the FactoMineR package [39].

**MSP database creation and validation.** The Bruker "MALDI Biotyper Preprocessing Standard Method" was used with slight modification namely the use of 2000 and 20,000 Da as lower and upper bound limits for spectra trimming. High quality spectra from fresh cercariae were selected for addition into the database using the Main Spectra Profile (MSP) creation tool of Compass Explorer using Bruker guidelines. Eight to 22 spectra were used for MSP creation.

Hierarchical Cluster Analysis (HCA) was performed using correlation method and the Ward algorithm for clustering with the MSP dendrogram tool of Compass Explorer.

Spectra used for MSP database validation came from *Schistosoma* cercariae from different emissions than those chosen for database construction. Both fresh and ethanol-stored cercariae were used.

According to our previous work on Trematoda's cercariae, identifications were considered interpretable when the first best match Log-Score Value (LSV1) was ≥ 1.7 (ROC curve presented on S1 Fig) [28]. Accuracy (percentage of correct identifications) and specificity were calculated in R (version 4.0.5) using the caret package [40].

For assessing the effects of various parameters on spectral quality and identification LSV1 value, spectra of hybrids were removed from the blind test validation dataset. Statistical analysis was performed in R (version 4.0.5) using base R with the rstatix, pROC, and forestmodel packages [41–43]. Student's t-test and Wilcoxon rank-test were used in univariate analysis; a logistic regression model was built for multivariate analysis. Bonferroni correction was used to adjust *p*-values in multiple comparisons.

**Differentiation between *Schistosoma haematobium* and Corsican hybrid strains using machine learning approach.** Four machine learning algorithms were evaluated to discriminate between *Schistosoma haematobium* and hybrid strains: k nearest neighbour (KNN), Support Vector Machine linear classifier (SVM), Partial Least-Square Discriminant Analysis (PLS) and Random Forest Analysis (RF) using custom R scripts based on the caret package [40].

Briefly, Bruker FID spectra files from *Schistosoma haematobium* and Corsican hybrid strains were imported using the MALDIQuantForeign package. Selection of high quality spectra was performed by using the screenSpectra function from the MALDIrppa package [44]. Spectra preprocessing and peak detection was performed with the MALDIquant package [45]. Intensity peak matrix was imported as a specmine object [46] and randomly split into "training" and "first validation" datasets. A second dataset containing both ethanol preserved and fresh specimens was created. Model training was performed on the "training" dataset with the "knn", "svmLinear", "pls" and "rf" caret [40] methods with repeated cross validation (10 folds and 10 repeats). ROC metrics were used to evaluate model performance.

## Results

### Database construction and MSP classification

5408 spectra were collected. Representative spectra are shown in Fig 1. Spectra were visually different between species and reproducible across replicates of the same species. Interestingly, spectra of F1 and F1' hybrids are visually different while spectra of *S. haematobium* and of the Corsican hybrid seem very similar.

For database construction, between 9 and 24 high-quality spectra were chosen to create MSP for each strains. They were added to the in-house Trematode's MSP database [28] (database structure is listed in S1 Table).

Cluster analysis showed a clear separation between *S. haematobium*, *S. bovis*, *S. mansoni* and *S. rodhaini* (Fig 2). All spectra obtained from specimens belonging to the same species are grouped together within the same cluster. Low heterogeneity is observed between MSP of the same species (distance level < 100 arbitrary units). Corsican hybrids spectra, however, are mixed with those of *S. haematobium*. *S. bovis* x *S. haematobium* hybrids form a cluster close to that of the *S. haematobium*/Corsican hybrid cluster, in which there is no grouping according to the directions of hybridization.

### Database validation

Among the 5408 spectra, 3277 were selected for database validation (2166 from fresh specimens and 1111 from ethanol stored specimens). The number of spectra and the species distribution according to experimental procedure is given in Table 1.

The choice of an LSV cut-off >1.7 for specific identification was supported by ROC-curve analysis (S1 Fig). We chose not to lower this score in order to possibly detect cercariae of non-schistosomal species [28].

1399 spectra (42.7%) had a LSV > 1.7. Among them, 80 (5.7%) non-concordant identifications were observed. The global accuracy of the test was 0.94 (95%CI [0.928, 0.953]). Specificity

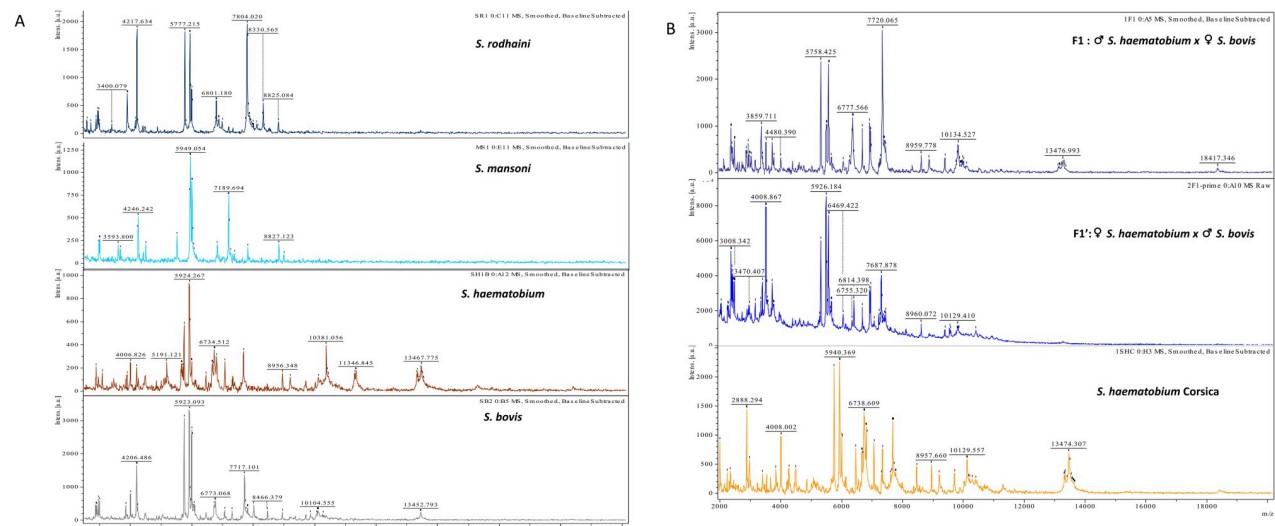

**Fig 1. Representative spectra of cercariae.** Panel A: The four species of *Schistosoma*: *S. rodhaini*, *S. mansoni*, *S. haematobium* and *S. bovis*. Panel B: *S. haematobium* x *S. bovis* F1 and F1' laboratory reared hybrids, *S. haematobium* x *S. bovis* Corsican hybrids. Smoothed spectra with baseline substracted. The m/z values are expressed in Da and the intensity of spectra are reported in arbitrary unit (a. u.).

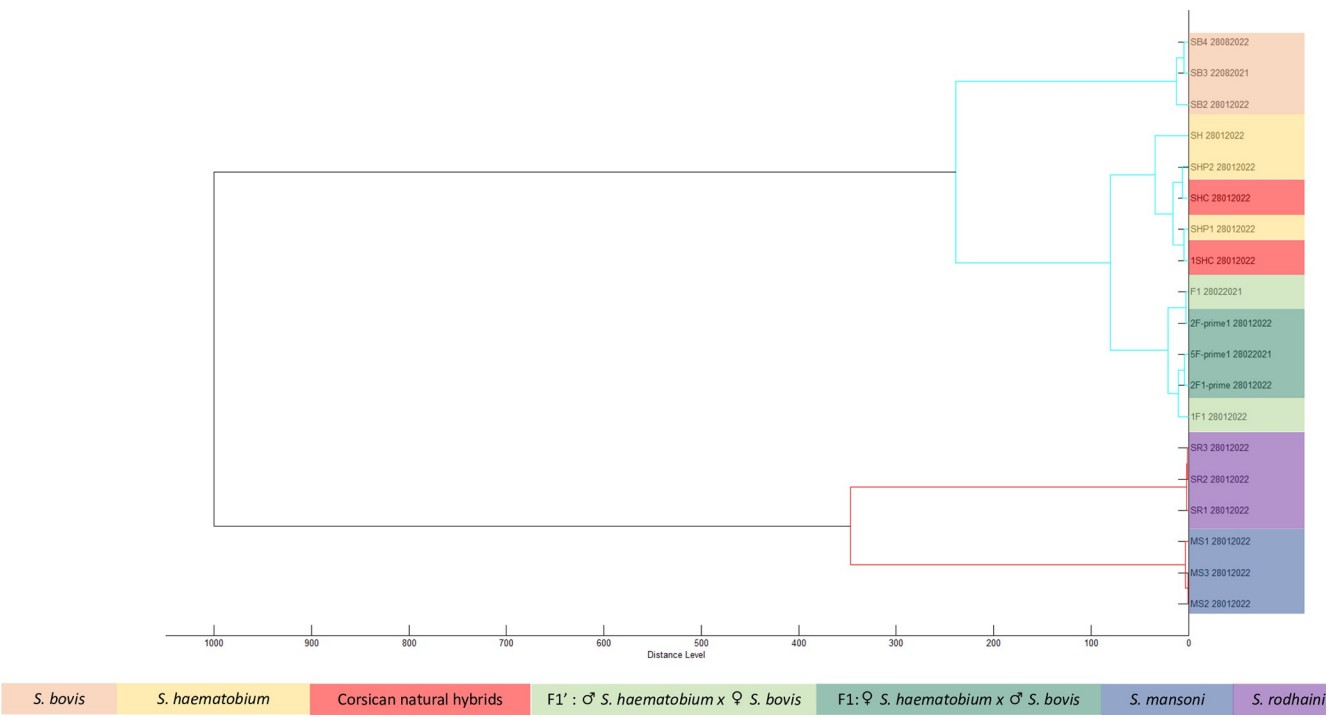

| S. bovis | S. haematobium | Corsican natural hybrids | F1' : ♂ *S. haematobium* x ♀ *S. bovis* | F1: ♀ *S. haematobium* x ♂ *S. bovis* | S. mansoni | S. rodhaini |

**Fig 2. Cluster analysis of MSP.** Spectra dendrogram was obtained using Pearson correlation coefficient and Ward D2 linking.

for identification of *Schistosoma* at the species level was good: *S. bovis*: 99.59%, *S. haematobium* 99.56%, *S. mansoni* and *S. rodhaini*: 100%.

Among discordant results, 68 *S. haematobium* spectra were wrongly identified as Corsican hybrids, 4 Corsican hybrids were misidentified as *S. haematobium*, 5 F1' hybrids were misidentified as *S. bovis* and one as *S. haematobium*. Lastly, one *S. bovis* was identified as an F1 hybrid with a LSV of 2.18.

## Factors impacting spectrum quality and species identification

The reproducibility of spectra was assessed by comparing spectra acquired from fresh specimens of cercariae belonging to the same strain and originating from different molluscs (Fig 3).

**Table 1. Number of spectra and species distribution.**

| Genotype | Number of spectra | | | |
| --- | --- | --- | --- | --- |
| | Database creation | Database validation | Quality test with LSV | Total number of acquired spectra |
| *Schistosoma bovis* Parental Strain | 65 | 401 | 401 | 570 |
| *Schistosoma haematobium* Parental Strain | 52 | 680 | 514[a] | 906 |
| *Schistosoma mansoni* Parental Strain | 64 | 740 | 740 | 1073 |
| *Schistosoma rodhaini* Parental Strain | 36 | 557 | 557 | 881 |
| F1 hybrids | 46 | 119 | 0 | 332 |
| F1' hybrids | 68 | 552 | 0 | 1078 |
| Corsican hybrid | 34 | 228 | 0 | 568 |
| **Total** | **365** | **3277** | **2212** | **5408** |

[a]166 spectra of *S. haematobium* Parental Strain were inadvertently acquired with the MBT_AutoX_smart method instead of MBT_BTS_Validation_AutoX and were thus excluded from the quality test with LSV analysis.

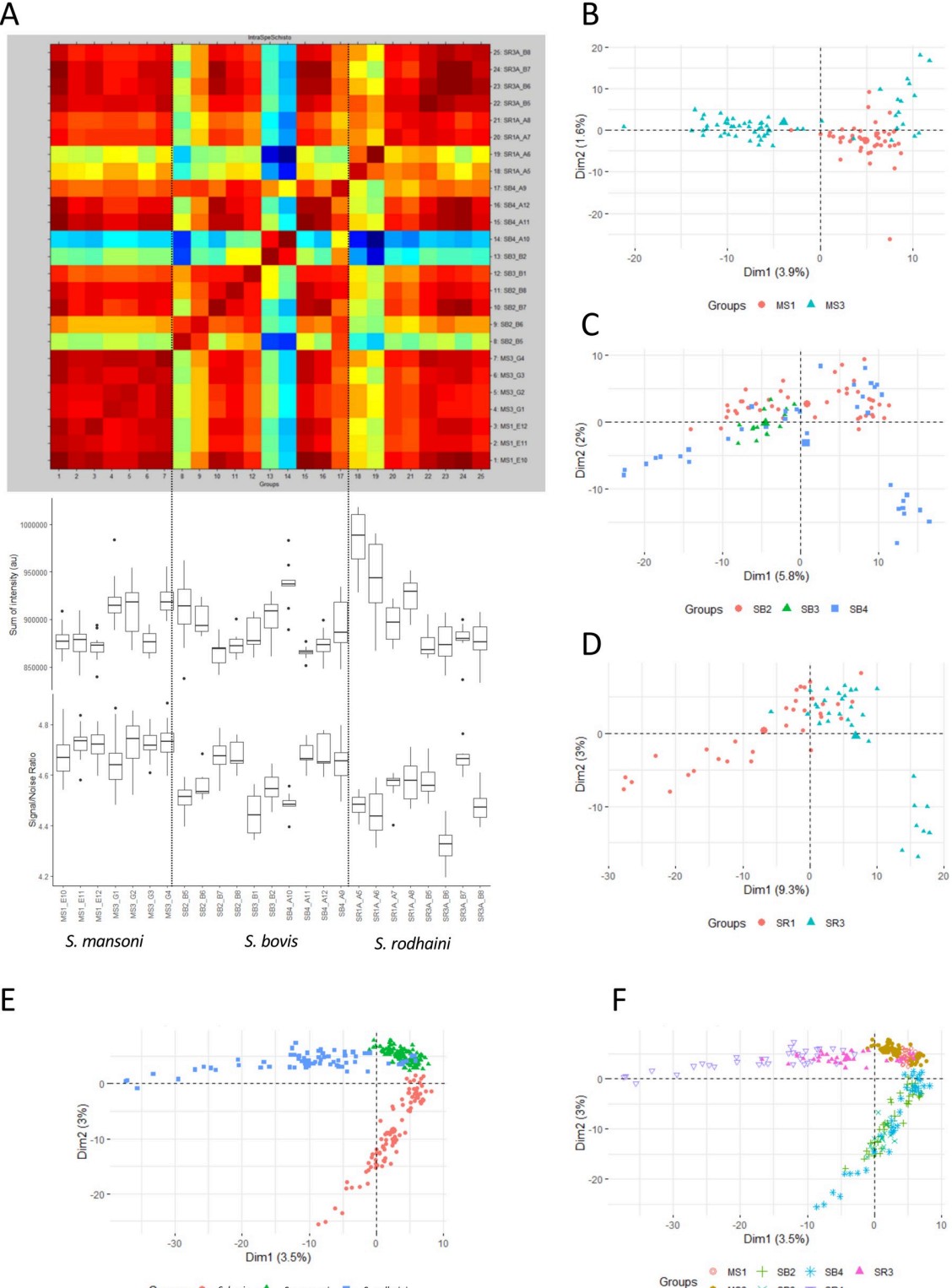

**Fig 3. Intraspecies spectra reproducibility.** Panel A. CCI matrix obtained with MALDI-Biotyper default parameters, sum of intensity and SNR for seven molluscs. B. First two principal components of PCA analysis of *S. mansoni* spectra. C. First two principal components of PCA analysis of *S. bovis* spectra. D First two principal components of PCA analysis of *S. rodhaini* spectra. E. First two principal components of PCA analysis of the whole reproducibility dataset, colours representing species. F. First two principal components of PCA analysis of the whole reproducibility dataset, colours representing species.

These analyses were performed for *S. bovis* (3 molluscs: SB2, SB3 and SB4), *S. mansoni* (2 molluscs MS1 and MS3) and *S. rodhaini* (2 molluscs SR1A, SR3A). The composite correlation index matrix (Fig 3A) showed high similarity between all spectra except for SB2_B5, SB3_B2 and SB4_A10. These spectra were characterized by a high intensity but a low signal-to-noise level. Manual inspections showed a peaks pattern similar to other spectra from the same species. All these spectra were unambiguously correctly identified at species level. Principal Component Analysis (PCA) of the MS1/MS3 spectra showed a dispersion of the MS3 spectra along Principal Component 1, while MS1 spectra were clustered to the positive side of this component (Fig 3B). For *S. bovis* (except for SB3) and *S. rodhaini*, spectra were dispersed along the PC1 and PC2 (Fig 3C and 3D). The analysis of the whole reproducibility dataset showed a clustering by species with an overlap between some *S. mansoni* and *S. rodhaini* spectra (Fig 3E). These overlapping spectra correspond to the SR3 mollusc (Fig 3F). In each case, only less than 15% of variance is explained by the first 2 components of PCA showing that a large part of the spectral diversity is not explained by the emitting snail or by the species to which the cercaria belongs.

In order to evaluate the factors influencing quality of spectra, we compared spectra of the same mollusc at day 0 (with and without ethanol fixation) and at day 22 (after ethanol fixation). As previously described [28], the intensity of spectra decreased significantly with ethanol fixation (mean 893889.8 vs. 865683.4 in case of ethanol fixation $p < 0.001$) Fig 4A. This difference was not significant between fresh specimens and day 22 fixed specimens (mean total intensity 893889.8 vs 865432.9 $p = 0.33$). The SNR slightly increase with ethanol fixation (mean SNR 4.54 vs 4.62 $p<0.001$) as shown in Fig 4B. These differences vary according to species (Fig 4C).

The influence of ethanol preservation, the number of deposited cercariae and species on identification success was then evaluated on a larger dataset: 2212 spectra were selected, 1101 from fresh specimens and 1111 from ethanol stored specimens (Quality test with LSV in Table 1).

Distribution of best match LSV (LSV1) according to the number of deposited cercariae is presented in Fig 5, for fresh and ethanol preserved specimens, according to the number of deposited cercariae (Fig 5A and 5B), or the species of *Schistosoma* (Fig 5C and 5D). LSV1 value were significantly lower (mean 1.54 vs 1.84, median 1.41 vs 2.08, $p < 0.0001$ Student t-test) for ethanol preserved specimens. 344 spectra out of 1111 (30.9%) from ethanol-preserved cercariae did, however, reach the threshold of 1.7.

For fresh specimens, LSV1 was significantly associated with the number of deposited cercariae (adjusted $p < 0.0001$, Wilcoxon rank test). Best results were obtained with 4–5 cercariae (mean LSV1 2.13, median 2.24, 77/86 spectra with LSV1 > 1.7), then for 2–3 cercariae (mean LSV1 1.98, median 2.22, 260/341 spectra with LSV1 > 1.7). Spectra with valid LSV1 were obtained in 319/478 (66.7%) spectra for only one cercaria (mean LSV1 1.73, median 1.97). When the number of deposited cercariae was superior to 5, we observed a degradation of LSV1 values, with only 9/60 (15%) spectra achieving LSV1 > 1.7 (mean LSV1: 1.50; median 1.48). When comparing LSV1 between groups, only "2–3 cercariae" and "4–5 cercariae" groups did not differ significantly.

For ethanol preserved cercariae, best LSV1 values were observed when 2–3 cercariae were deposited (mean LSV1: 1.9, median: 2.16, 140/233 spectra with LSV1 > 1.7).

For both ethanol and fresh specimens, the species of *Schistosoma* is another factor associated with LSV1 (adjusted $p < 0.001$) with best LSV1 values observed for fresh specimens of *S. bovis* (mean: 2.1, median: 2.21 and 42/152 spectra with LSV > 1.7).

In order to identify a factor associated with obtaining a valid LSV1 (> 1.7), a logistic regression model was built for spectra with LSV > 0 (Fig 6). In this multivariate analysis, the factors affecting the probability of LSV1 > 1.7 were ethanol preservation (odds-ratio of 0.10 CI95% [0.07–0.14], $p < 0.001$), the number of cercariae (odds-ratio for 2–3 cercariae of 139.26 CI95%

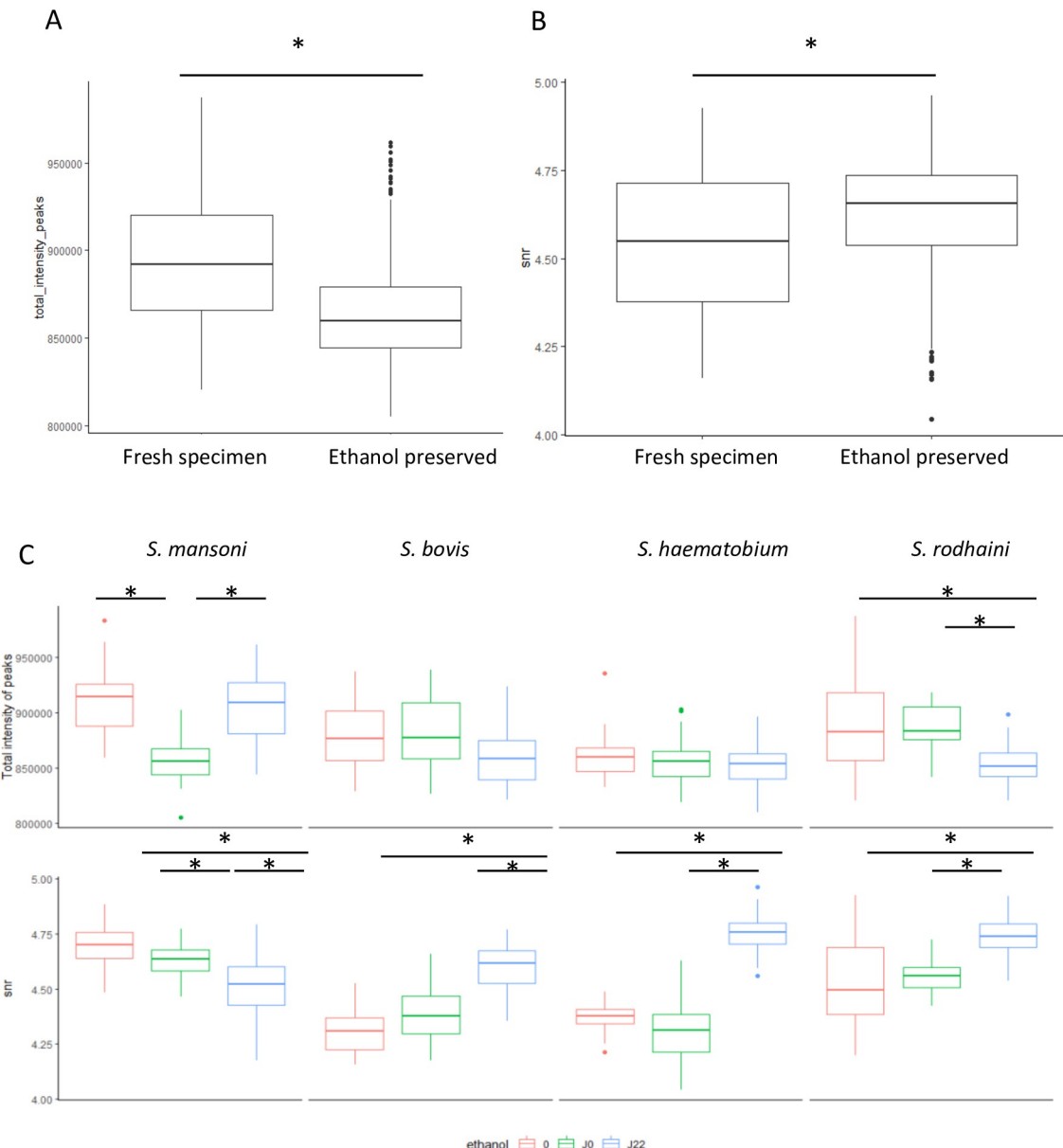

**Fig 4. Effect of ethanol fixation on spectral features.** A. Boxplot of the sum of peaks intensities in fresh and ethanol preserved specimens. B. Boxplot of the SNR in fresh and ethanol preserved specimens. C. Boxplots of the sum of peaks intensities and SNR in fresh and ethanol preserved specimens according to species. * Statistically significant difference (p adjusted by Bonferroni correction).

[64.14–333.1], p < 0.001 Spectrum acquired from *S. haematobium* cercariae have a lower probability of obtaining LSV1 >1.7 (odds-ratio of 0.40 CI95% [0.27–0.58]) than those acquired from *S. mansoni* cercariae (odds-ratio of 4.54, CI95% [3.15, 6.57], p < 0.001).

## Machine learning for discrimination of the Corsican hybrid

Most misidentifications come from confusion between *S. haematobium* and the Corsican hybrids. Machine learning was thus tested in order to unambiguously distinguish the two species. A dataset containing spectra (n = 649) from fresh cercariae of pure *S. haematobium*

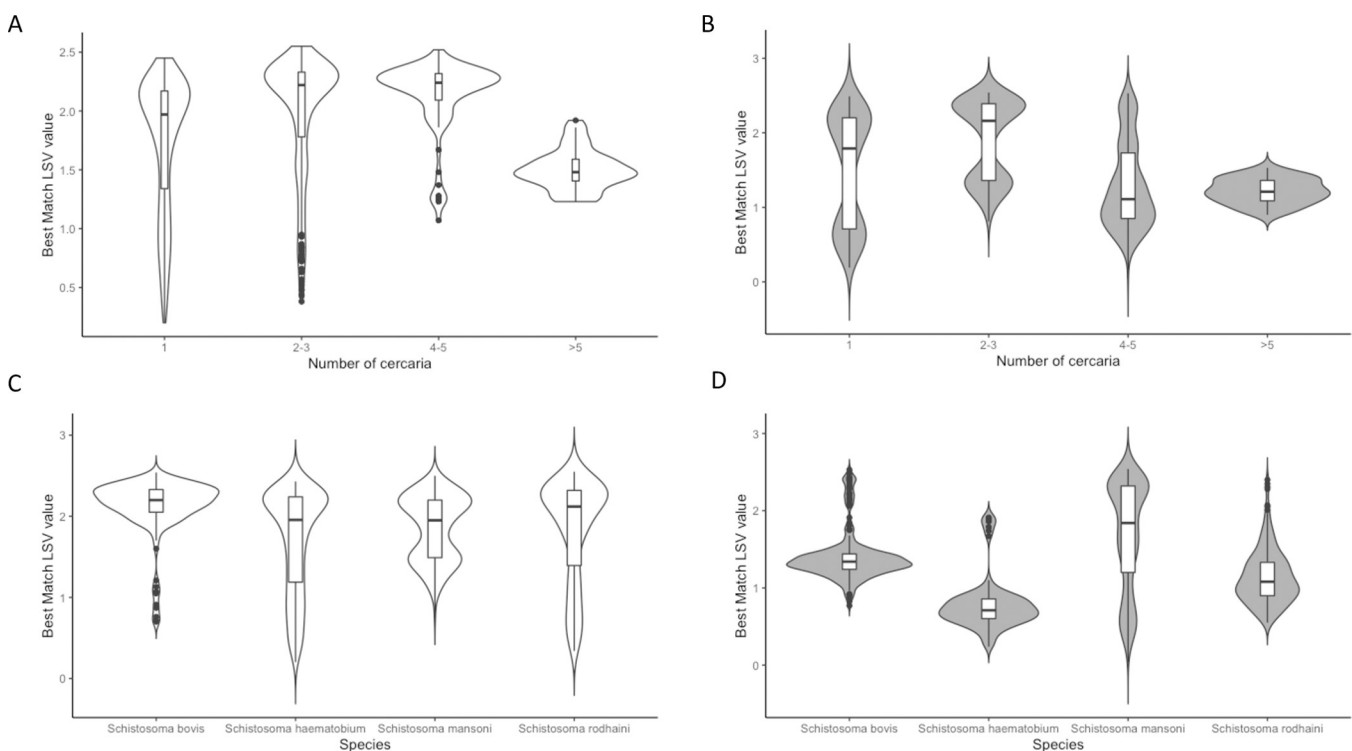

**Fig 5. Influence of ethanol preservation, number of deposited cercariae and species on LSV.** Violins plot of best match LSV value for: A. Fresh specimens according to the number of deposited spectra. B. Ethanol preserved specimens according to the number of deposited spectra. C. Fresh specimens according to the species. D. Ethanol preserved specimens according to species.

(n = 335) and Corsican hybrids (n = 312) deposited on 5 plates and acquired over five different days, has been selected. This dataset was then splitted into a training dataset (344 spectra: *S. haematobium* n = 172, Corsican hybrid n = 172) and validation dataset (303 spectra: *S. haematobium* n = 140, Corsican hybrids n = 163). A second validation dataset (116 spectra: *S. haematobium*, ethanol preserved n = 20, Corsican hybrids n = 96) was also selected. All models achieved a good classification performance on the first validation dataset with accuracy, F1 score and Sensitivity/Specificity > 97% (Table 2).

The second dataset containing both ethanol preserved and fresh specimens displayed a performance drop for the SVM, PLS and RF models while the KNN model kept a very good accuracy (0.9914), sensitivity (0.9896) and specificity (1.000).

## Discussion

MALDI-TOF is a reliable and accurate tool for the identification of *Schistosoma* cercariae of medical importance at the species level. We confirm here the potential of MALDI-TOF for a fast screening of Trematode's cercariae, a capacity that could be developed for the environmental monitoring of human and animal schistosomiasis outbreaks.

Inter-species hybridization occurs naturally in the genus *Schistosoma*, notably within *S. haematobium* and *S. bovis*, whose introgressed population are responsible of urogenital schistosomiasis outbreaks in West Africa and Corsica [47]. The use of MALDI-TOF to identify *S. haematobium* x *S. bovis* hybrids seems promising. Spectra from F1 laboratory reared hybrids with 50% of each parental genome [35] clustered distinctly from spectra of parental *S. haematobium* strain and the Corsican hybrid strain. They can be identified with high specificity

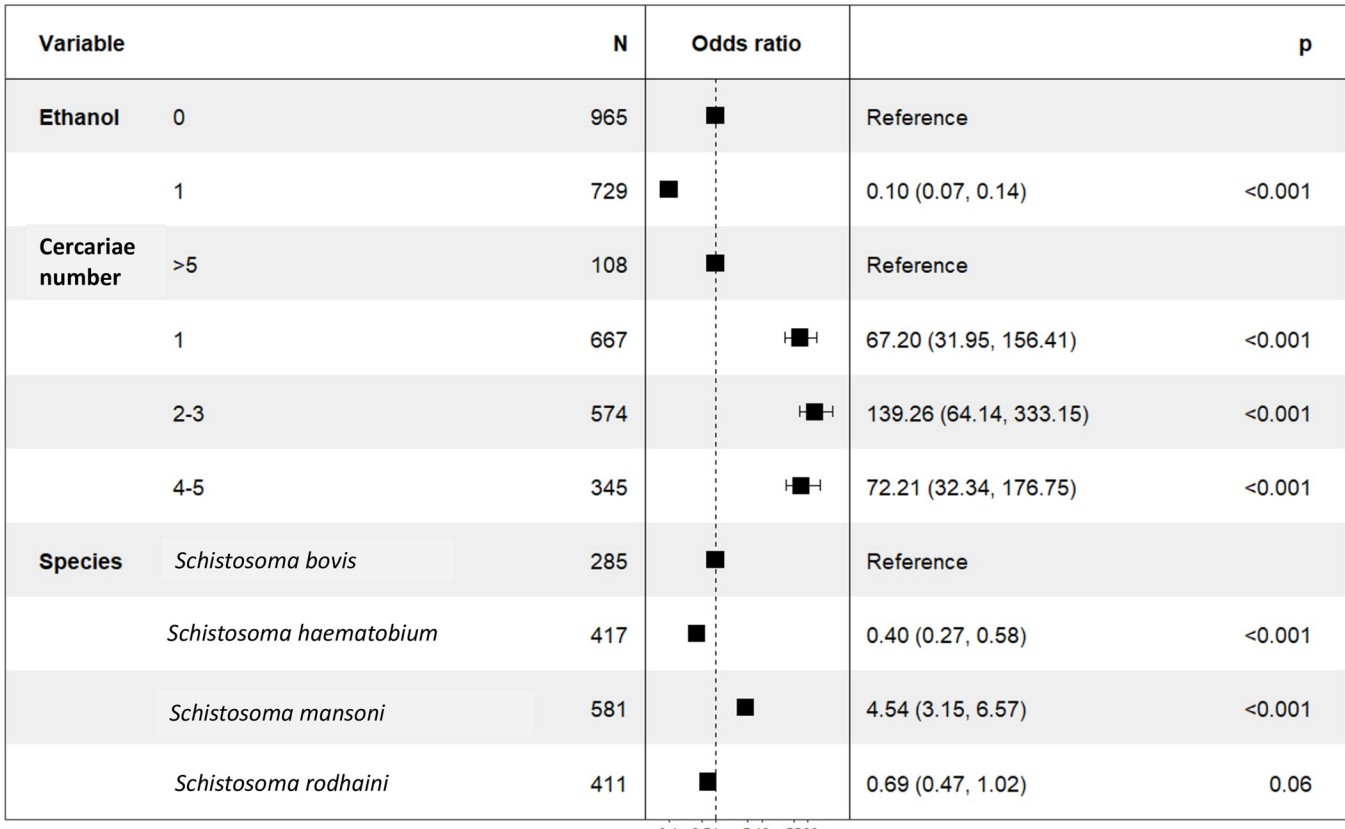

**Fig 6. Multivariate analysis of factors influencing spectral identification.** Forest plot of significant variables used in logistic regression model for prediction of LSV1 > 1.7.

using MALDI-TOF. Spectra of the Corsican hybrids are grouped together with those of the Egyptian parental *S. haematobium* strain, indicating close spectra proximity. This result is coherent with the high level of introgression in the Corsican hybrid genome as 77% is of *S. haematobium* origin and 23% of *S. bovis* origin [48]. Corsican hybrids spectra, however, can be distinguished from those of the Egyptian *S. haematobium* strains with the identification algorithm used routinely (global accuracy ~94%) and more robustly using machine learning techniques with an accuracy > 99%.

**Table 2. Performance of various machine learning models for the detection of Corsican hybrids on first and second validation datasets.**

| Model | Validation dataset | TP | TN | FN | FP | Accuracy | F1 score | Sensitivity | Specificity | PPV | NPV |
|---|---|---|---|---|---|---|---|---|---|---|---|
| KNN | First dataset | 163 | 138 | 0 | 2 | 0.9934 | 0.9939 | 1.000 | 0.9857 | 0.9879 | 1.000 |
|  | Second dataset | 95 | 20 | 1 | 0 | 0.9914 | 0.9948 | 0.9896 | 1.0000 | 1.0000 | 0.9524 |
| SVM | First dataset | 159 | 139 | 4 | 1 | 0.9835 | 0.9845 | 0.9755 | 0.9929 | 0.9937 | 0.9720 |
|  | Second dataset | 88 | 20 | 8 | 0 | 0.931 | 0.9565 | 0.9167 | 1.0000 | 1.0000 | 0.9565 |
| PLS | First dataset | 160 | 140 | 3 | 0 | 0.9901 | 0.9907 | 0.9816 | 1.0000 | 1.0000 | 0.9790 |
|  | Second dataset | 89 | 20 | 7 | 0 | 0.9397 | 0.9622 | 0.9271 | 1.0000 | 1.0000 | 0.7407 |
| RF | First dataset | 159 | 140 | 4 | 0 | 0.9868 | 0.9876 | 0.9755 | 1.0000 | 1.0000 | 0.9722 |
|  | Second dataset | 80 | 19 | 16 | 1 | 0.8534 | 0.9040 | 0.8333 | 0.9500 | 0.9877 | 0.5429 |

TP: True Positives; TN: True Negatives; FN: False negatives; FP: False Positives; PPV: Positive Predictive Value; NPV: Negative Predictive Value

Based on parasite specimens sampled in several African countries, it has been evidenced that hybrids may harbour different patterns of genomic introgression across *S. haematobium* lineages [47]. It will thus be interesting to evaluate the ability of MALDI-TOF to detect hybrids with various introgression rates, ranging from low admixture to complete genomic introgression of nuclear genomes derived from *S. haematobium* or *S. bovis*. The sensitivity of MALDI-TOF could also be tested to diagnose the diversity of hybrid genotypes at the population level (i.e. from various transmission sites). Hybrids of other *Schistosoma* species, including *S. haematobium* x *S. guineensis* and *S. haematobium* x *S. mansoni*, may also be added and tested.

As in our previous study, we noted a low ratio between acquired spectra and spectra reaching LSV1 cut-off (~40%). However, the repetition of the number of acquisitions per well and of the number of deposits allows one to obtain a satisfactory identification rate per snail. Several factors affecting the quality of the spectra acquired and the probability of obtaining a valid identification score were investigated. The univariate and multivariate analysis confirmed a significant performance drop for samples fixed with ethanol. The influence of ethanol on the MALDI-TOF spectra varies according to the studied micro-organism: for arthropods, freezing seems to be a better conservation method than ethanol [49]. A "de-alcoholization" protocol for analysing ticks preserved in ethanol was proposed [50]. Hamlili et al. reported that ethanol storage of snails significantly modify spectra profiles and decrease the identification performances despite the use of de-alcoholization protocol [29].

Concerning helminths, only one paper by Wendel et al. [51] has compared the different storage media for the identification of *Taenia saginata* proglottid's. No degradation of spectra quality was observed for proglottids stored at -20˚C in 70% ethanol for 24 weeks. However this methodology needs to be reproduced with other species and life-stages of helminths. In our previous study on the application of MALDI-TOF on identification of Trematoda's cercariae we already noticed a decrease of spectra intensity and identification success with ethanol fixation [28]. Nevertheless, identification remains possible for approximately 30% of the spectra, which makes it worth trying to attempt the study of specimens preserved in alcohol. It would be necessary to evaluate other means of preservation of cercariae in order to facilitate retrospective studies. Alternatively, the identification of specimens preserved in ethanol could be improved by building a specific database as proposed by Hamlili et al. [29] or by using machine learning to search for robust peaks allowing the identification of degraded spectra.

The number of cercariae deposited is a critical factor for the success of the identification, with the best results obtained by depositing 2 to 5 cercariae per well. This study also demonstrates the ability of obtaining a spectrum from a single cercaria. The quantity of biological material brought by a cercaria is thus sufficient to obtain a spectrum, the lowest success rate being probably linked to the random process of MALDI-TOF laser shots on the target spot, ensuring the ionization. The use of smaller spots can potentially increase the success rate by reducing the number of "missed" shots and would allow more specimens to be analysed in the same target plate, thus increasing the speed of analysis and reducing the cost per sample. The acquisition of the individual spectrum of cercaria is particularly interesting for studying coinfections within the same snail. Indeed, in a recent survey conducted in Senegal, 15/88 snails were reported as shedding cercariae from two or more species of *Schistosoma*, suggesting a high rate of co-infection [10].

The WHO strategy for the eradication of human schistosomiasis relies heavily on mass anthelmintic drug administration campaigns. Although it could efficiently reduce the morbidity linked to this infection, this strategy does not offer protection against reinfection and thus remains problem-prone for long-term eradication of schistosomiasis, especially in areas of high prevalence [52–54]. On the other hand, control of snails (by molluscicides, destruction of habitat or biological control) is an highly effective way to durably reduce schistosomiasis

prevalence [55–57]. In order to target and monitor snails-control programs, large malacological surveys are needed. These surveys involve the screening of a large number of snails and identification of released cercariae. Classic morphological assessment of both snails and cercariae are a difficult and laborious task which is not suitable for large studies. Environmental DNA has been developed to identify the presence of either *Schistosoma* single species using a qPCR approach [13,58] or Trematoda communities using a metabarcoding approach [59]. To date, these methods cannot discriminate hybrid Schistosomes. Indeed, the qPCR approach targets the Dra1 nuclear sequence which is present in both *S. bovis* and *S. haematobium*; and the meta-barcoding approach targets a mitochondrial (16S) uni-parentally inherited marker. Moreover, because these methods detect the parasite population and not single individuals it is not possible to differentiate between the presence of hybrids and the simultaneous presence of two species. Image analysis by conventional neural networks has also recently been proposed for the identification of snails and cercariae from photos taken in the field [11]. This approach is promising but currently limited by the number of species present in the model. Moreover, it is not suitable for hybrid identification due to the absence of morphological differences. Whatever the method used, eDNA, Image analysis, and MALDI-TOF, it is crucial to improve reference banks.

Hamlili et al. [29], have demonstrated that MALDI-TOF is a reliable tool for the rapid identification of frozen and ethanol-stored freshwater snails intermediate hosts of schistosomiasis, permitting to distinguish intermediate hosts and non-intermediate hosts without any malacological expertise.

In the present study, we confirm the usefulness of MALDI-TOF for cercariae identification, with a great potential for malacological surveys in *Schistosoma*-endemic areas, enabling the rapid and reliable identification of emitted cercariae at species level and detection of hybrids. Identification of the snails could also be obtain with MALDI-TOF using the database containing nine species published by Hamlili et al. [29], making it a tool able to identify both the host and the parasite.

One of the limitations of MALDI-TOF is its inability to detect cercariae directly from snail's tissues, making it impossible to identify pre- or non-patent infection [29]. However, the majority of pre-patent snails never realised their cercariae [60]. Their importance in transmission is probably weak, and interpretation of their epidemiological role is difficult.

Another important limitation of this work is the low accessibility of MALDI-TOF in developing countries endemic for schistosomiasis. However MALDI-TOF has been proven particularly useful for clinical diagnostic and epidemiology in the "Senegalese Network For The Exploration Of Non-malarial Causes Of Fever" [61]. We hope that access to MALDI-TOF will expand in tropical areas, but at the moment use of MALDI-TOF for a field malacological investigation will raise logistical problems for the transport of specimens to a reference laboratory, in the endemic country or abroad. In this context, a robust, inexpensive preservation technique that does not require refrigerated transport would be particularly useful.

## Conclusion

MALDI-TOF is a reliable technique for high-throughput identification of Schistosome cercariae of medical and veterinary importance, even in case of hybridization. The spectral base remains to be completed to allow the identification of more human and animal species.

## Supporting information

**S1 Table. Composition of spectral database.**
(DOCX)

**S1 Fig. ROC plot of LSV Value for specific identification of cercariae in the blind-test dataset.**
(PDF)

## Acknowledgments

We thank Eva Nast for her technical support, Aline Cuénod for accepting reuse of her R script and Matthieu Kaltenbach for proofreading the manuscript.

## Author Contributions

**Conceptualization:** Antoine Huguenin, Jérôme Depaquit, Jérôme Boissier, Hubert Ferté.

**Methodology:** Antoine Huguenin, Jérôme Boissier, Hubert Ferté.

**Software:** Antoine Huguenin.

**Supervision:** Jérôme Depaquit, Hubert Ferté.

**Validation:** Antoine Huguenin, Julien Kincaid-Smith.

**Visualization:** Antoine Huguenin.

**Writing – original draft:** Antoine Huguenin.

**Writing – review & editing:** Antoine Huguenin, Julien Kincaid-Smith, Jérôme Depaquit, Jérôme Boissier, Hubert Ferté.

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
