## [Decision Letter · Decision Letter 0]

27 Aug 2022

Dear Dr Huguenin,

Thank you very much for submitting your manuscript "MALDI-TOF : A new tool for the identification of Schistosoma cercariae and detection of hybrids" for consideration at PLOS Neglected Tropical Diseases. As with all papers reviewed by the journal, your manuscript was reviewed by members of the editorial board and by several independent reviewers. In light of the reviews (below this email), we would like to invite the resubmission of a significantly-revised version that takes into account the reviewers' comments. 

We cannot make any decision about publication until we have seen the revised manuscript and your response to the reviewers' comments. Your revised manuscript is also likely to be sent to reviewers for further evaluation.

Sincerely,

Shan Lv

Section Editor

Reviewer's Responses to Questions

**Key Review Criteria Required for Acceptance?**

**Methods**

-Are the objectives of the study clearly articulated with a clear testable hypothesis stated?

-Is the study design appropriate to address the stated objectives?

-Is the population clearly described and appropriate for the hypothesis being tested?

-Is the sample size sufficient to ensure adequate power to address the hypothesis being tested?

-Were correct statistical analysis used to support conclusions?

-Are there concerns about ethical or regulatory requirements being met?

Reviewer #1: The study is very interesting but need more solid data.

Reviewer #2: Please find my detailed report below.

**Results**

-Does the analysis presented match the analysis plan?

-Are the results clearly and completely presented?

-Are the figures (Tables, Images) of sufficient quality for clarity?

Reviewer #1: Yes, the analysis is good.

Reviewer #2: Please find my detailed report below.

**Conclusions**

-Are the conclusions supported by the data presented?

-Are the limitations of analysis clearly described?

-Do the authors discuss how these data can be helpful to advance our understanding of the topic under study?

-Is public health relevance addressed?

Reviewer #1: The data should be emphasized more.

Reviewer #2: Please find my detailed report below.

**Editorial and Data Presentation Modifications?**

Reviewer #1: (No Response)

Reviewer #2: -Please find my detailed report below.

**Summary and General Comments**

Reviewer #1: The result showed the big difference between the different species of parasite in Fig 1. Some peaks above m/z 10000 in Schistosoma haematobium and bovis, Does the author check that and get the peptide sequence?

All the worm parasites were collected from the host in laboratory, could you please evaluate the detection mode with the wide type of cercaria from the field?

The author described that “Molecular biology techniques allow a reliable diagnosis but are expensive. TOF is a recent technique that permits an inexpensive”, but I think the MALDI-TOF is not inexpensive but high throughput, so it maybe not expensive if you can solve the problem of sample size.

The study of this paper is interesting and seems promising, the accuracy of the learning machine is good for detection, however, it could not detect the snail, which is important for the policy of schistosomiasis control and it may waste too much time for the parasite grown to the stage of cercaria. 

MALDI-TOF technique has been applied in the early detection of schistosomiasis, some reference should be noticed with PMID 31166908. 

Some minor error:

Spell error for the word of schistosomiasis in Author Summary.

Reviewer #2: This is an interesting manuscript that reports research on an innovative idea, i.e. the application of MALDI-TOF mass spectrometry, a common routine diagnostic method for identification of bacteria and fungi in the microbiology lab, for identification of cercariae stemming from Schistosoma. The authors found that a reliable identification was achieved for most different cercariae, with exception of hybrids between S. haematobium and S. bovis, which were ‘misclassified’ as S. haematobium. The identification could be further improved using artificial intelligence algorithms.

I have some observations that might be addressed by the authors:

1) Abstract: Please spell out MALDI-TOF at first mention.

2) Introduction: Well written chapter. Line 100: The official name of Ivory Coast is Côte d’Ivoire (should not be translated), please correct.

3) Introduction, line 109: Please define MALDI-TOF at first mention. Also, I would not describe it as a “new technique” for bacteria identification as it has been used for almost 15 years in this context. 

4) Methods: Accurate description of the work, well written. Lines 210-212: In bacteriology, log score values (LSVs) >1.7 are usually considered as genus-specific and LSVs >2.0 as species-specific. Even though it remains unclear whether this applies likewise to parasites, it would be useful/desirable to report identification rates for both established cut-offs in the Results section (and perhaps in the Abstract). This should also be mentioned more clearly in the Methods and/or Discussion, because in the current form, LSVs >1.7 are repeatedly described as species-specific, e.g. in the first sentence of the Discussion. This should be modified, as it might be misleading.

5) Discussion, line 417: typo error, please correct to S. guineensis.

6) Discussion, lines 423-430: The significant influence of ethanol on spectra quality is a little surprising, as it differs from previous papers on MALDI-TOF for parasites, e.g. PubMed ID 34683327. Please mention and discuss this issue.

7) Discussion: In the limitations of the work, it should be mentioned that many small- and medium-sized laboratories in the tropics do not (yet) have access to MALDI-TOF machines.

PLOS authors have the option to publish the peer review history of their article (what does this mean?). If published, this will include your full peer review and any attached files.

Reviewer #1: No

Reviewer #2: No
---

## [Editor Report · Decision Letter 1]

29 Nov 2022

Dear Dr Huguenin,

Thank you very much for submitting your manuscript "MALDI-TOF : A new tool for the identification of Schistosoma cercariae and detection of hybrids" for consideration at PLOS Neglected Tropical Diseases. As with all papers reviewed by the journal, your manuscript was reviewed by members of the editorial board and by several independent reviewers. In light of the reviews (below this email), we would like to invite the resubmission of a significantly-revised version that takes into account the reviewers' comments. 

We cannot make any decision about publication until we have seen the revised manuscript and your response to the reviewers' comments. Your revised manuscript is also likely to be sent to reviewers for further evaluation.

Sincerely,

Shan Lv, Ph.D.

Section Editor

Shan Lv

Section Editor
---

## [Decision Letter · Decision Letter 2]

6 Mar 2023

Dear Dr. Huguenin,

We are pleased to inform you that your manuscript 'MALDI-TOF : A new tool for the identification of Schistosoma cercariae and detection of hybrids' has been provisionally accepted for publication in PLOS Neglected Tropical Diseases.

Best regards,

Eva Clark, M.D., Ph.D.

Section Editor

Eva Clark

Section Editor

Reviewer's Responses to Questions

**Key Review Criteria Required for Acceptance?**

**Methods**

-Are the objectives of the study clearly articulated with a clear testable hypothesis stated?

-Is the study design appropriate to address the stated objectives?

-Is the population clearly described and appropriate for the hypothesis being tested?

-Is the sample size sufficient to ensure adequate power to address the hypothesis being tested?

-Were correct statistical analysis used to support conclusions?

-Are there concerns about ethical or regulatory requirements being met?

Reviewer #1: (No Response)

Reviewer #2: Yes, adequate.

**Results**

-Does the analysis presented match the analysis plan?

-Are the results clearly and completely presented?

-Are the figures (Tables, Images) of sufficient quality for clarity?

Reviewer #1: (No Response)

Reviewer #2: Yes, adequate. Specifically, all my comments pertaining to the use of ethanol as fixative have been addressed in detail.

**Conclusions**

-Are the conclusions supported by the data presented?

-Are the limitations of analysis clearly described?

-Do the authors discuss how these data can be helpful to advance our understanding of the topic under study?

-Is public health relevance addressed?

Reviewer #1: (No Response)

Reviewer #2: Yes

**Editorial and Data Presentation Modifications?**

Reviewer #1: (No Response)

Reviewer #2: -

**Summary and General Comments**

Reviewer #1: (No Response)

Reviewer #2: All my previous comments and suggestions have been addressed sufficiently.

PLOS authors have the option to publish the peer review history of their article (what does this mean?). If published, this will include your full peer review and any attached files.

Reviewer #1: No

Reviewer #2: **Yes: **Soeren L Becker

---

## [Editor Report · Acceptance letter]

22 Mar 2023

Dear Dr. Huguenin,

We are delighted to inform you that your manuscript, "MALDI-TOF : A new tool for the identification of Schistosoma cercariae and detection of hybrids," has been formally accepted for publication in PLOS Neglected Tropical Diseases.

Best regards,

Shaden Kamhawi

co-Editor-in-Chief

Paul Brindley

co-Editor-in-Chief
